# Multi-scale Sinusoidal Embeddings Enable Learning on High Resolution Mass Spectrometry Data

## Abstract

Small molecules in biological samples are studied to provide information about disease states, environmental toxins, natural product drug discovery, and many other applications. The primary window into the composition of small molecule mixtures is tandem mass spectrometry (MS2), which produces high sensitivity and part per million resolution data. We adopt multi-scale sinusoidal embeddings of the mass data in MS2 designed to meet the challenge of learning from the full resolution of MS2 data. Using these embeddings, we provide a new state of the art model for spectral library search, the standard task for initial evaluation of MS2 data. We also investigate the task of chemical property prediction from MS2 data, that has natural applications in high-throughput MS2 experiments and show that an average $R^2$ of 80% for novel compounds can be achieved across 10 chemical properties prioritized by medicinal chemists. We vary the resolution of the input spectra directly by using different floating point representations of the MS2 data, and show that the resulting sinusoidal embeddings are able to learn from high resolution portion of the input MS2 data. We apply dimensionality reduction to the embeddings that result from different resolution input masses to show the essential role multi-scale sinusoidal embeddings play in learning from MS2 data.

## 1 Introduction

Metabolomics is the study of the small molecule ($\lesssim 1{,}000\,\mathrm{Daltons}$) contents of complex biological samples. Tandem Mass Spectrometry (MS/MS), in conjunction with chromatography, is one of the most commonly used tools in metabolomics. Tandem Mass Spectrometry works by measuring with very high resolution the masses of molecules and their constituent fragments. While MS/MS techniques are highly sensitive and precise, inferring the identity of the molecules and their properties from the resulting mass spectra is commonly regarded as one of metabolomics' primary bottlenecks (Dunn et al., 2013). Improved tools for these tasks will impact applications across many areas of science including disease diagnostics, characterization of disease pathways, development of new agrochemicals, improved forensics analysis, and the discovery of new drugs (Zhang et al., 2020).

Profiling unknown molecules with mass spectrometry consists of several steps. First, molecules of interest are ionized and separated by their mass to charge ratio ($m/z$), resulting in the MS1 spectrum. Then, individual "precursor" ions are fragmented, and the $m/z$'s of the fragments are recorded in the same manner. The resulting spectrum contains the $m/z$'s and intensities (together, the "peaks") of all resulting fragments, and is called the MS2 spectrum. See Glish & Vachet (2003).

In recent years, several machine learning methods have been developed to identify the structures and properties of small molecules from their mass spectra. These approaches (Huber et al., 2021a;b; Kutuzova et al., 2021; Litsa et al., 2021; Shrivastava et al., 2021; van Der Hooft et al., 2016) historically discretize $m/z$ (via tokenization or binning). However, the $m/z$ values obtained in modern mass spectrometry experiments are collected with parts per million levels of resolution. There are three critical reasons to expect that modeling $m/z$ values as numeric quantities, in contrast to discretized values, is the appropriate technique. First, we know that relevant chemical information is present at the millidalton level (Jones et al., 2004; Pourshahian & Limbach, 2008) and discretization schemes typically strip this information away. Additionally, mass differences between peaks represent

fragmentation patterns, and are therefore relevant to understanding the molecule. Learning to utilize this information from tokens is exceedingly difficult. Finally, discretization is susceptible to edge effects, where slight mass difference can map to different bins if the masses are close to the edge of a bin. In light of these considerations, we model $m/z$ as a numerical value using sinusoidal embeddings, which we hypothesize will enable us to capture information across many orders of magnitude.

In this work, we apply a numerical representation of $m/z$ that uses sinusoidal embeddings across multiple scales to retain the information content of MS2 data across its entire mass resolution. To demonstrate the ability of these embeddings to enable learning at the high resolution of MS2 experimental data, we apply them to a search task and 10 regression tasks.

Our first test task is spectral library search (Stein & Scott, 1994). In spectral library search, spectra from unknown compounds are compared to spectra in a database to find matches using a similarity function over pairs of spectra. This task is the primary method used in standard metabolomics analyses, but is challenging because spectra for a compound vary widely with experimental conditions. We find that a similarity function based on sinusoidal embeddings achieves state of the art both for finding the exactly correct compound, and also for finding close structural analogs, which is useful for compounds not contained in spectral databases.

We further investigate predicting chemical properties relevant to drug discovery from MS2 data and apply the same modeling approach to this task. We achieve 80% average $R^2$ for out of sample molecules across 10 properties, which is high enough to enable first-pass filtering and selection of candidate drug molecules in high-throughput experiments based solely on spectral data.

In each task, using sinusoidal embeddings results in a new state of the art. The confirmation of across-task improvement provides evidence that the embeddings are a general improvement, rather than task specific. To determine whether these results are due to learning from the high resolution portion of the data, we experiment with inputting MS2 data cast to half precision floating point numbers, and show that the performance noticeably degrades relative to double precision. Finally, we visualize embeddings generated with varied floating point precision MS2 inputs using UMAP (McInnes et al., 2018) projections and show that non-trivial high dimensional structure only emerges with sufficiently high precision input. Taken together, these results are the first clear evidence that sinusoidal embeddings enable effective learning from high mass resolution MS2 data across tasks in metabolomics.

## 1.1 RELATED WORK

Modeling numerical data in terms of sinusoidal functions has a long history in many scientific fields. In machine learning, sinusoidal embeddings are most commonly used to encode the discrete positions of natural language token inputs in transformer models, which are otherwise position-agnostic (Vaswani et al., 2017). Other work has used sinusoidal embeddings for multi-dimensional positional encoding in image recognition (Li et al., 2021a). In mass spectrometry, sinusoidal embeddings have been used in proteomics for inferring protein sequences from mass values, either with (Qiao et al., 2019) or without (Yilmaz et al., 2022) an initial mass binning step, but without exploring their role in model performance or comparing to alternative approaches. These techniques have never been applied in the domain of metabolomics. Metabolomics differs from proteomics in that the molecules of of interest are 2 - 3 orders of magnitude lower mass and are graph structured rather than sequential as in proteins. Consequently, the tasks and challenges of modeling of small molecules are sharply divergent from those in proteomics.

For modeling $m/z$ values in the field of metabolomics, previous machine learning models have relied primarily on discretization of the continuous mass inputs. This is usually accomplished by binning $m/z$ values into fixed length vectors with peak intensity as the value for each element. Various authors have used binned representations of spectra for spectral library search (Huber et al., 2021b), unsupervised topic modeling (van Der Hooft et al., 2016), and molecule identification (Kutuzova et al., 2021; Litsa et al., 2021). Alternatively, masses have been tokenized via rounding for tasks such as unsupervised spectral similarity (Huber et al., 2021a) and molecule prediction from synthetic data (Shrivastava et al., 2021).

Other approaches rely on tokenization of $m/z$ values by assigning a molecular formula to each peak, and taking the formula as a token. Even when this method is able to uniquely identify molecular formulas for every fragment ion, it still faces a problem endogenous to tokenization: it's very difficult to reason about the relationships between discrete tokens except by pattern recognition over a very large number of training examples. Böcker & Rasche (2008) addressed this difficulty by modeling the fragmentation process with a tree structure, and Dührkop et al. (2015) used these fragmentation trees as inputs to a molecular fingerprint prediction model that is used to assign molecules to spectra from molecular structure databases. The tree-based approaches are prohibitively computationally expensive for large volumes of data Cao et al. (2021).

## 2 MODEL

### 2.1 BASE MODEL ARCHITECTURE

An MS2 spectrum,

$$S = \left\{ (m/z, I)_{\text{precursor}}, (m/z, I)_{\text{fragment}_1}, \ldots, (m/z, I)_{\text{fragment}_N} \right\}, \tag{1}$$

is composed of a precursor $m/z$ and a set of $N$ fragment peaks at various $m/z$'s and intensities ($I$). Various other data are typically collected, including precursor abundance, charge, collision energy, *etc*, but these are not used in this work. Transformer encoders (Vaswani et al., 2017) without the positional encoding are explicitly fully-symmetric functions and hence are ideally suited to model a set of fragmentation peaks. We therefore take our base model, $\text{SpectrumEncoder}(S)$, to be a transformer encoder, whose inputs are a set of $(m/z, I)$ pairs that includes the precursor along with all of the fragment peaks. We normalize the intensities to a maximum of 1 for fragments and assign an intensity of 2 to the precursor. Finally, we take as output the embedding vector from the final transformer layer corresponding to the precursor input. See Figure 2. Note, the use of transformers to study synthetic MS2 spectra has been explored in Shrivastava et al. (2021).

This flexible approach enables us to experiment with and compare various representations of MS2 data. We pursue two approaches: tokenization of $m/z$ and modeling $m/z$ as numerical values via sinusoidal embeddings. Before describing these, we need the following definition for a simple two layer feed forward MLP,

$$\text{FF}(x) = W_2 \text{ReLu}(W_1 x + b_1) + b_2. \tag{2}$$

Note that each occurrence of FF we employ below is a separate instance with distinct weights.

#### 2.1.1 TOKENIZED $m/z$ PEAK EMBEDDING

Our first approach is to discretize $m/z$ by rounding to $0.1$ precision and treating these objects as tokens (as in an NLP context). These tokens are embedded in a dense vector space as in Huber et al. (2021a); Mikolov et al. (2013), via an embedding function TE. We then construct the token peak embedding,

$$\text{PE}_{\text{token}}(m/z, I) = \text{FF}(\text{TE}(m/z) \parallel I), \tag{3}$$

where $\parallel$ denotes concatenation. A diagram of $\text{PE}_{\text{token}}$ is shown in Figure 1a.

#### 2.1.2 SINUSOIDAL $m/z$ PEAK EMBEDDING

We now consider a numerical peak embedding that utilizes the full $m/z$ precision. We employ a sinusoidal embedding as follows:

$$\text{SE}(m/z, 2i; d) = \sin\left(2\pi \left[\lambda_{\min}\left(\frac{\lambda_{\max}}{\lambda_{\min}}\right)^{2i/(d-2)}\right]^{-1} m/z\right) \tag{4}$$

$$\text{SE}(m/z, 2i+1; d) = \cos\left(2\pi \left[\lambda_{\min}\left(\frac{\lambda_{\max}}{\lambda_{\min}}\right)^{2i/(d-2)}\right]^{-1} m/z\right). \tag{5}$$

The frequencies are chosen so that the wavelengths are log-spaced from $\lambda_{\min} = 10^{-2.5}$ Daltons to $\lambda_{\max} = 10^{3.3}$ Daltons, corresponding to the mass scales we wish to resolve. This embedding

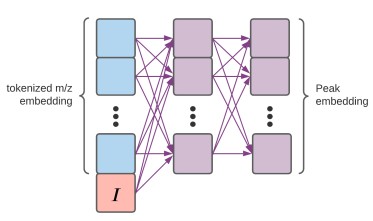
(a) Architecture of $\text{PE}_{\text{token}}$.

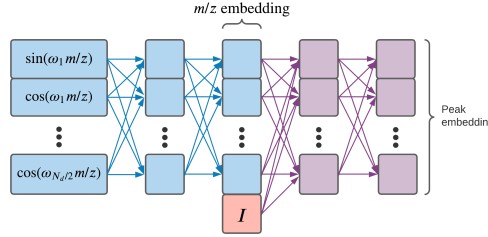
(b) Architecture of $\text{PE}_{\text{sin}}$.

Figure 1: Architectures used to embed MS2 peaks (composed of a $m/z$ and an intensity $I$). Both approaches produce an embedding of $m/z$ of dimension $d$. The $m/z$ embedding in Figure 1a (left) is produced by a learned embedding layer. The $m/z$ embedding in Figure 1b (right) is produced by a sinusoidal embedding followed by a two layer MLP. In both cases, an intensity value $I$ is appended to make a vector of dimension $d + 1$. This is then fed into a simple feed-forward network to produce a peak embedding of dimension $d$. We use blue to highlight the network components involved in producing an $m/z$ embedding and purple to highlight the full peak embedding after intensity is incorporated.

is inspired by Li et al. (2021a); Vaswani et al. (2017). Similarly to Equation 3, we construct the sinusoidal peak embedding,

$$\text{PE}_{\text{sin}}\left(m/z, I\right) = \text{FF}\left(\text{FF}\left(\text{SE}\left(m/z\right)\right) \,\|\, I\right). \tag{6}$$

We also experiment with casting the $m/z$ inputs as half, single, and double precision floating point values. Equations 4-5 are computed at the precision specified by $m/z$ and the results are then cast to the precision of the model weights used for training. A diagram of $\text{PE}_{\text{sin}}$ is shown in Figure 1b.

### 2.1.3 TRANSFORMER

The set of fragment peaks and the precursor are passed through either of the two peak embeddings specified above. The result is a sequence of embedding vectors which is then passed through a transformer encoder:

$$\text{SpectrumEncoder}\left(S\right) =$$
$$\text{TransformerEncoder}\left(\text{PE}\left(m/z, I\right)_{\text{precursor}}, \ldots, \text{PE}\left(m/z, I\right)_{\text{fragment}_N}\right). \tag{7}$$

Here PE stands in for either $\text{PE}_{\text{token}}$ or $\text{PE}_{\text{sin}}$. TransformerEncoder has embedding dimension $d = 512$ and six layers, each with 32 attention heads and an inner hidden dimension of $d$. Note that all hidden and embedding layers in our peak embeddings above have dimension $d$ as well. As mass spectra have no intrinsic ordering, we opt to not include a positional encoding.

In order to get a single embedding vector as an output, the final transformer layer query only attends to the first embedding, corresponding to the position of the precursor $m/z$. A diagram of our full model architecture is shown in Figure 2.

### 2.2 MODEL TRAINING

We train the base models discussed here via two independent tasks aimed at applications within metabolomics and medicinal chemistry. These tasks are described in detail below. We build these models using PyTorch (Falcon, 2019; Paszke et al., 2019) and train them with the Adam (Kingma & Ba, 2014) optimization algorithm with parameters $\beta_1 = 0.9$, $\beta_2 = 0.999$, and learning rate $\alpha = 5.0 \times 10^{-5}$. We also use a weight decay (Krogh & Hertz, 1991) parameter of $0.1$, a gradient clipping a parameter of $0.5$, and a dropout (Srivastava et al., 2014) parameter of $0.1$. All models were trained using between 25 and 50 epochs. Finally, all model weights were trained using half precision floating point values, regardless of the precision of the $m/z$ inputs.

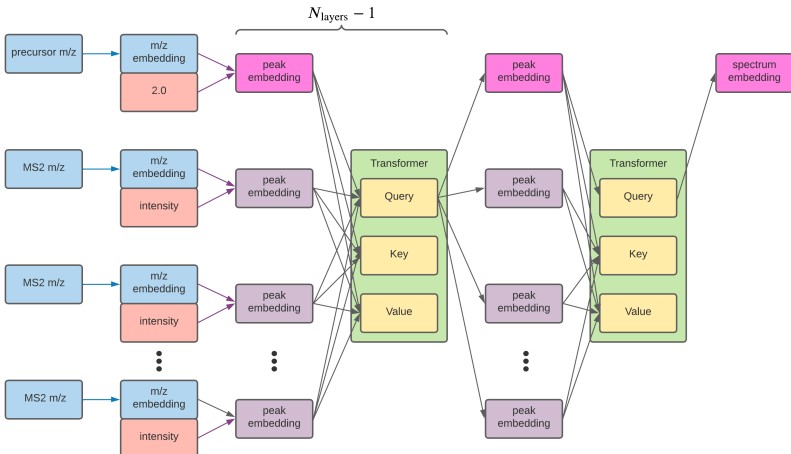

Figure 2: Full model architecture that highlights how we featurize and embed an MS2 spectrum in a dense vector space.

### 2.2.1 SPECTRAL SIMILARITY WITH SIAMESE NETWORKS

A common workflow for the analysis of mass spectrometry data is to compute some notion of spectral similarity for pairs of spectra that is intended to correlate with the molecular similarity of the underlying compounds (Yilmaz et al., 2017). This can then be used for tasks such as spectral library search (Stein & Scott, 1994) and molecular networking (Quinn et al., 2017; Watrous et al., 2012). Historically, most spectral similarity metrics used in the field have been heuristic-based (Li et al., 2021b; Yilmaz et al., 2017). More recently, spectral similarity methods that make use of techniques from deep learning have been developed (Huber et al., 2021a;b). The latter of these, MS2Deepscore, is a Siamese network that trains a simple MLP (by binning $m/z$) to predict a measure of molecular similarity from pairs of mass spectra. As is common practice in this field, we use a molecular similarity given by a Tanimoto similarity computed from RDKit Landrum et al. (2013) topological fingerprints of the corresponding molecular structures (Bajusz et al., 2015). The loss function is the mean square error ($MSE$) between the ground truth molecular similarity and the cosine similarity evaluated on the dense embedding vectors generated by the base model.

Molecular similarity scores range from 0 to 1. The scores of randomly sampled pairs from our labeled dataset are not uniformly distributed and strong similarity matches are rare. Huber et al. (2021b), describe a procedure to sample pairs of spectra such that the molecular similarities will be uniformly distributed. We use this procedure to sample pairs for training and evaluation tasks.

Following Huber et al. (2021b) we also train a Siamese network, but we do so using the base model transformer architectures outlined in Section 2.1 instead of with the binned mass vectors used by Huber. We report performance results for models trained using $\mathrm{PE}_{\mathrm{token}}$ and $\mathrm{PE}_{\mathrm{sin}}$ below in Section 3.2.

### 2.2.2 PROPERTY PREDICTION

We also train the base model architecture to predict 10 chemical properties relevant to medicinal chemistry directly from an MS2 spectrum. We do so by training the following network,

$$\text{Properties} = \text{FF}\left(\text{SpectrumEncoder}\left(S\right)\right), \qquad (8)$$

where the final layer has dimension equal to the number of properties we wish to predict. We also scale all of our training labels to zero mean and unit variance. For inference, we run predicted properties through an inverse-scaler. As in Section 2.2.1, we train multiple versions of this property prediction model. Due to the lack of existing methods for comparison, we also train a version of Equation 8 on binned spectrum representations for a simple baseline. We replace SpectrumEncoder with the MS2Deepscore feed forward architecture found in Huber et al. (2021b) for this baseline.

The properties we predict are standard indicators of druglikeness and bioavailability in the field of medicinal chemistry. These include the properties that make up the Quantitative Estimate of Druglikeness (QED) along with several others (Bickerton et al., 2012; Lipinski et al., 2012; Veber et al., 2002). For the complete list of properties, see Table 2. All properties are computed deterministically from chemical structure using RDKit (Landrum et al., 2013), so there is no additional dependency on experimental data.

## 3 RESULTS

### 3.1 DATA

We construct a set of labeled MS2 spectra by combining a number of standard public and commercial datasets and one small dataset ( 0.1% of spectra) that requires permission to access because it contains psychoactive substances (Wang et al., 2016; Mehta, 2020; Sawada et al., 2012; Horai et al., 2010; Mikaia et al., 2014; Smith et al., 2005; Mardal et al., 2019). See our Reproducibility Statement for additional preprocessing details. Chemical properties and fingerprints are then computed from the cleaned molecules using RDKit (Landrum et al., 2013). Our resulting dataset has 1,251,830 spectra corresponding to 45,351 distinct structures.

In metabolomics experiments, biological samples contain compounds that have previously been profiled in MS2 libraries ("known" compounds) and compounds that have not been profiled ("novel compounds"). Identifying the structure and properties of known molecules is an important task on its own (*e.g.* "dereplication"). We therefore split our data so that the test and development sets are partially disjoint from train at the level of structures, and fully disjoint from train at the level of spectra. This enables us to evaluate model performance on both "known" and "novel" compounds. To implement this split, we partition our set of labeled spectra into a training set, development set (*i.e.* validation set), and test set as follows. We randomly select a set of 1002 molecules from our data and place them and all of their associated spectra into the development set. From the remaining molecules, we follow the same procedure and place 1002 molecules into the test set. This creates a test and development set containing 1002 molecules each that are fully disjoint from each other. The remaining molecules are assigned the training set. We then remove 998 spectra that correspond to 941 molecules represented in the training set to the development set, and 998 to the test set, corresponding to 953 structures from the training set. The final training, development, and test sets have 1,214,812, 18,750, and 18,268 spectra respectively. The training set contains 43,347 structures. We use the development set for hyperparameter optimization and only access the test set to compute reported metrics.

### 3.2 SPECTRAL SIMILARITY

To benchmark spectral library search, we apply two criteria for accurate molecular identification: retrieving the exact molecule and retrieving a molecule which is an approximate match, *i.e.* has molecular similarity (defined in section 2.2.1) greater than 0.6. To account for widely varying numbers of spectra for each molecule, we report the accuracies as macro averages over molecular structures.

We train a number of spectral similarity models that report state of the art performance or are in standard usage as described in Section 2.2.1. We refer to the sinusoidal Siamese transformer and tokenized Siamese transformer models as the sinusoidal and tokenized models respectively. To test performance against reported state of the art models, we train both a Spec2Vec (Huber et al., 2021a) and an MS2Deepscore (Huber et al., 2021b) model on our data. We also show results modified cosine spectral similarity, which is a standard, unlearned, similarity function common in applications (Watrous et al., 2012). In Table 1 we report spectral library search accuracies for both known and novel compounds obtained from a number of spectral similarity models. We omit exact match accuracies for novel compounds since these have been sampled so that no exact matches can be found. For models where it is applicable, the same table reports the $MSE$ loss between predicted and actual molecular similarity on spectrum pairs drawn from train, known, and novel spectra sets.

We find that learned approaches improve on modified-cosine spectral library search. Of the learned approaches, we find that Spec2Vec has strong, only surpassed by the double precision sinusoidal model, spectral library search performance on novel, but only beats MS2Deepscore on known.

Table 1: Spectral library search performance

| Spectral Set Match | train | $MSE$ known | novel | Spectral library search accuracy | | |
| | | | | known | | novel |
| | | | | Exact | Approx. | Approx. |
|---|---|---|---|---|---|---|
| Modified Cosine | | | | 0.61 | 0.672 | 0.351 |
| Spec2Vec | | | | 0.903 | 0.954 | 0.414 |
| MS2Deepscore | 0.026 | 0.029 | 0.047 | 0.852 | 0.918 | 0.387 |
| Tokenized $m/z$ | 0.017 | 0.019 | 0.030 | 0.933 | 0.961 | 0.382 |
| Sinusoidal $m/z$ (`float16`) | 0.019 | 0.019 | 0.03 | 0.913 | 0.955 | 0.401 |
| Sinusoidal $m/z$ (`float64`) | **0.013** | **0.015** | **0.024** | **0.937** | **0.97** | **0.435** |

MS2Deepscore only outperforms the tokenized model on novel and underperforms everything on known. Moreover, MS2Deepscore is substantially less accurate at predicting the specific pair molecular similarity on known and novel spectra when compared to all transformer approaches. The double precision sinusoidal model produced the best performance on all evaluation tasks. This model produces state of the art known spectral library search accuracies of 0.937 and 0.97 for exact and approximate matching respectively. Numerical representation of MS2 $m/z$ via sinusoidal embeddings consistently outperforms all the discretization based approaches we benchmarked.

To assess whether the multi-scale embeddings are learning from high resolution information, we also train and evaluate versions of our sinusoidal model where $m/z$'s are cast to half precision floating point values. Half precision floating point can only resolve up to approximately parts per $\mathcal{O}(10,000)$, far below the parts per million precision of experimental mass spectrometry data. The performance of the half precision model, reported in Table 1, drops to be on par with the tokenized model (worse on known and better on novel). We do not report results with single precision $m/z$ since these were indistinguishable from double precision $m/z$. On all evaluation metrics, using $m/z$ inputs of precision greater than 16 bits dramatically improves performance of the sinusoidal model. This indicates that sinusoidal embedding of $m/z$ values enables learning from the the high resolution portion of mass spectrometry data.

### 3.2.1 QUALITATIVE EMBEDDING ANALYSIS

To further characterize the respective properties of token and multi-scale sinusoidal embeddings, we inspect UMAP (McInnes et al., 2018) projections of our $m/z$ embeddings in Figure 3. For this analysis, we use siamese transformer models described in Section 2.2.1. For our sinusoidal models, we embed $50,000$ $m/z$ values between 0 and $1,000$ Daltons using FF (SE $(m/z)$). We do not use the full peak embedding function as we are not interested in the embedding of intensity information. Because our tokenization procedure involves rounding $m/z$ values to 1 decimal place, for the tokenized model we only embed $10,000$ $m/z$ values between 0 and $1,000$ Daltons.

While $m/z$ measurements are continuous, the space they represent is inherently discrete. Two molecular fragments differing in atomic composition could nevertheless have a very similar $m/z$'s. However, the large separation between molecular mass scales and the mass scale at which nuclear forces manifest themselves means that the relative atomic composition may be deduced from the fractional $m/z$ (Jones et al., 2004; Pourshahian & Limbach, 2008), defined by $\{m/z\} = m/z - \lfloor m/z \rfloor$. Therefore, we expect a quality embedding of $m/z$ to embed fragments similar in both $m/z$ and $\{m/z\}$ close to one another, while paying greater attention to the latter.

As is seen in Figure 3, embeddings from tokenized and low precision sinusoidal models are able to capture general trends in $m/z$. However, they fail to preserve distances in $m/z$ and show little to no structure in $\{m/z\}$. In contrast, the high precision sinusoidal embeddings allow our models to represent important information in $\{m/z\}$, while preserving distance in $m/z$.

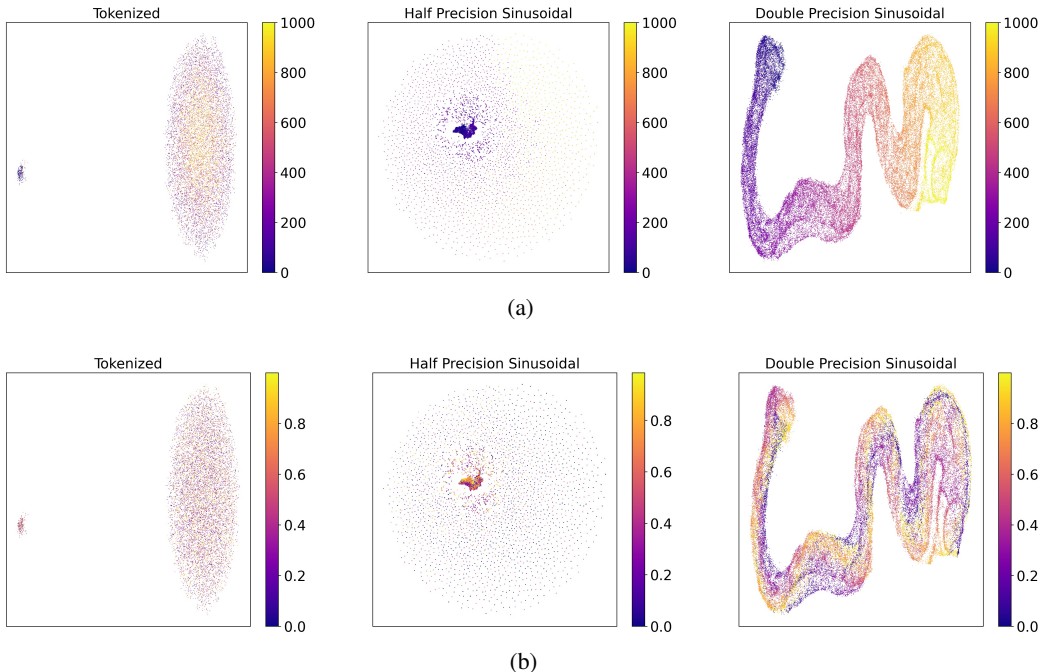

Figure 3: (3a) UMAP projections of $m/z$ embeddings colored by $m/z$ value. (3b) UMAP projections of $m/z$ embeddings colored by $\{m/z\}$. High resolution sinusoidal embeddings imbue model latent space with additional structure not found in other baseline models.

### 3.3 PROPERTY PREDICTION

Many applications in metabolomics can be unblocked with knowledge of just a limited set of chemical properties, without need for identification of molecular structure. Towards this end, we investigate the parallel task of chemical property prediction from MS2 spectra. We evaluate sinusoidal *vs.* tokenized $m/z$ embedding performance on a list of properties that have compelling applications in drug discovery and are easily computable from our molecule labels. For this experiment, we train a sinusoidal model and several baseline models as described in Section 2.2.2.

In Table 2, property prediction models are evaluated on known and novel structures using $R^2$. All transformer models outperform our feed forward baseline. Half precision sinusoidal models produced an $R^2$ of 0.746 (0.936) on known (novel) molecules averaged across all 10 properties. These results are competitive with tokenization but substantially underperform when compared to higher precision sinusoidal embeddings. When $m/z$ values are specified at double precision, sinusoidal embeddings show improvements over tokenization on each property, resulting in an 11% improvement over tokenization on $R^2$ for novel molecules averaged across all properties.

This is further evidence that the multi-scale information is captured by sinusoidal embeddings and allows improved generalization across tasks. In addition, the performance is strong enough to enable medicinal chemistry applications, such as hit prioritization based on druglikeness. This has major implications in the drug discovery context, as inspecting the properties of unknown molecules in complex mixtures was previously not possible without difficult isolation steps and individual experimentation.

## 4 CONCLUSIONS

The results presented here are the first demonstration that the sinusoidal representation of $m/z$s measured in tandem mass spectrometry experiments enables deep learning models to learn from high resolution mass data. These results include state of the art performance on spectral library search and

Table 2: $R^2$ for predicted chemical properties

| Property | Feed Forward | | Tokenized $m/z$ | | Sinusoidal $m/z$ | |
|---|---|---|---|---|---|---|
| | Known | Novel | Known | Novel | Known | Novel |
| all | 0.824 | 0.604 | 0.943 | 0.72 | **0.976** | **0.8** |
| atomic $\log P$ | 0.768 | 0.357 | 0.919 | 0.437 | **0.968** | **0.622** |
| number of hydrogen bond acceptors | 0.835 | 0.762 | 0.969 | 0.887 | **0.987** | **0.941** |
| number of hydrogen bond donors | 0.834 | 0.572 | 0.924 | 0.683 | **0.968** | **0.69** |
| polar surface area | 0.851 | 0.716 | 0.965 | 0.862 | **0.986** | **0.907** |
| number of rotatable bonds | 0.801 | 0.669 | 0.939 | 0.792 | **0.976** | **0.838** |
| number of aromatic rings | 0.838 | 0.372 | 0.934 | 0.474 | **0.981** | **0.655** |
| number of aliphatic rings | 0.839 | 0.638 | 0.943 | 0.762 | **0.974** | **0.821** |
| number of heteroatoms | 0.823 | 0.719 | 0.968 | 0.902 | **0.988** | **0.946** |
| fraction of sp3 carbons | 0.846 | 0.618 | 0.951 | 0.723 | **0.977** | **0.821** |
| quantitative estimate of druglikeness | 0.806 | 0.617 | 0.912 | 0.681 | **0.952** | **0.755** |

property prediction tasks in metabolomics. The property prediction results of are sufficient quality to functionally inform and advance drug-discovery efforts.

Moreover, we present results that compare across numerical floating point precisions for the mass inputs, and show that sinusoidal embeddings perform better when higher mass resolution data is used. Finally, we visualize high-dimensional structure of the mass embeddings that emerges only when high precision mass values are utilized. We expect that sinusoidal embeddings of $m/z$ will be a useful component of further machine learning applications with MS2 data.

## REPRODUCIBILITY STATEMENT

We train our model on a combination of free public datasets (Wang et al., 2016; Mehta, 2020; Sawada et al., 2012; Horai et al., 2010), commercially available datasets (Mikaia et al., 2014; Smith et al., 2005), and one small dataset (0.1% of all spectra) available if a user-group of qualified researchers is joined (Mardal et al., 2019). The necessary preprocessing steps are outlined in Section 3.1. Additionally, we allow spectra collected across instrument types and experimental parameters including positive and negative ion mode, collision energies, etc. We constrain that spectra in our dataset have at least 5 peaks and at least 3 decimal places of $m/z$ resolution. We strip all stereochemistry from our molecular structure labels, which is a common step taken in MS2 modeling and allows for better molecule-disjoint splitting.

The model architecture is explained in detail in Section 2 and all parameters used in model training are specified in Section 2.2. We include our source code in the supplementary materials.

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
