# OpenReview forum: "Multi-scale Sinusoidal Embeddings Enable Learning on High Resolution Mass Spectrometry Data"
_ICLR.cc/2023/Conference — Submitted to ICLR 2023_

### Official Review · Reviewer_4K3S · 2022-10-16

**Confidence:** 3
**Correctness:** 3
**Technical Novelty And Significance:** 2
**Empirical Novelty And Significance:** 2
**Recommendation:** 5

**Clarity, Quality, Novelty And Reproducibility:**

I think the lack of reproducibility in both the dataset and implementation aspects hinders the impact and quality of this paper.

**Strength And Weaknesses:**

Strength: The idea of sinusoidal embeddings seems to be well-motivated for this special task: MS analysis. Also, the authors explain the idea quite clearly and the paper is well-written.

Weakness: Firstly, I think the analysis of tandem mass spectrometry is a very niche problem in metabolomics, which seems to be less relevant to the focus of the ICLR conference. I was wondering if this paper could submit to a more specialized conference and journal.
Second, it seems to the reviewer that neither the code nor the dataset can be made publicly available because of intellectual property reasons, which are against the spirit of ICLR and the reviewer doubts its potential impact on the scientific community.

**Summary Of The Paper:**

In this paper, the authors proposed multi-scale sinusoidal embeddings for spectral library search.  Then the proposed sinusoidal m/z peak embeddings are fed into a transformer to get a single embedding vector as an output, which is then fed into a feedforward neural network to predict chemical properties.

**Summary Of The Review:**

Overall, because of the niche topic that is less relevant to ICLR and the closed-source dataset and code, I don't think this paper is appropriate for ICLR acceptance.

---

> ### Author Response · Authors · 2022-11-19
> **Response to reviewer 4K3S**
>
> *Strength: The idea of sinusoidal embeddings seems to be well-motivated for this special task: MS analysis. Also, the authors explain the idea quite clearly and the paper is well-written.*
>
> We thank the reviewer for their comments. We want to emphasize that in addition to showing that sinusoidal embeddings provide a performance boost across tasks, we showed that they only provide the full boost when provided with high resolution inputs. This implies that they are capturing high mass resolution structure to learn.
>
> *Weakness: Firstly, I think the analysis of tandem mass spectrometry is a very niche problem in metabolomics, which seems to be less relevant to the focus of the ICLR conference. I was wondering if this paper could submit to a more specialized conference and journal.*
>
> While this is a fair question, we disagree with the characterization of MS/MS as a niche problem in metabolomics. Rather, in conjunction with chromatography it is the primary experimental technique in the field. Indeed, one of our citations discusses survey results that indicate that identifying chemical structures from MS/MS data is the primary bottleneck in metabolomics. So, far from being a niche problem, it is a candidate for the outstanding computational problem in the field. We hope that researchers at ICLR being exposed to this problem will engage more research into this critically important problem with implications across human health, environmental science, basic biology research, and beyond. We have attempted to clarify this point in the revised manuscript.
>
> *Second, it seems to the reviewer that neither the code nor the dataset can be made publicly available because of intellectual property reasons, which are against the spirit of ICLR and the reviewer doubts its potential impact on the scientific community.*
>
> We have added the code as a supplement for review, and will release the code upon publication. We have removed the tiny amount of proprietary in-house data from our datasets. Unfortunately, as noted in our response to another reviewer, substantial portions of the data are both standard in the field and only available commercially. It would be a violation of our license to release them, but we believe it is valuable to the community to see what results look like over datasets that include standard sources like NIST that are only available with license. We have cited all our sources, and note that many mass spectrometry labs will already be in possession of these datasets.

---

### Official Review · Reviewer_oTst · 2022-10-18

**Confidence:** 4
**Correctness:** 3
**Technical Novelty And Significance:** 2
**Empirical Novelty And Significance:** 2
**Recommendation:** 5

**Clarity, Quality, Novelty And Reproducibility:**

I covered these points above.


**Strength And Weaknesses:**

Overall, the paper makes a modest contribution in terms of novelty, and with some reorganization could tell a reasonably compelling story.

One problem is that the paper is not written in a way that is accessible to an ICLR audience.  I will try to outline below some places where I got confused, but keep in mind that I have a lot more experience working with mass spec data than your typical ICLR reader.  I'm a bit worried that you will find my list of questions below frustrating to read, because probably many of the answers to my questions are embedded in the text.  My goal is to show you how a reasonably well-informed reader got confused by the paper as written.

In the paragraph that spans pp. 1-2, I don't understand why discretization techniques should be "particularly susceptible to measurement errors."  Also, it seems strange to me that this discussion doesn't mention edge effects, which is another clear drawback of the discretization approach (separate from the loss of precision).

The final paragraph of the introduction does not do a good job of making clear whether the proposed method represents an improvement relative to the current state of the art.

There is something important that I simply don't understand about the spectral library search application.  My understanding is that the state of the art method is to use a sort of generalized scalar product (or cosine similarity) that is carried out directly on the non-discretized list of peaks.  I think this is the method used in several widely used library search tools, e.g., from Pieter Dorrenstein's group.  I think the paper should explain how methods like the ones cited in Section 1.1 (e.g., Huber 2021b) improve on this simple approach that does not require discretization.  Results from a simple method like this, as well as perhaps cosine similarity carried out on discretized vectors, should be included in Table 1.

Also in Section 1.1, please explain what it means to do discretization by "direct molecular formula assignment."  Without a definition, it's impossible to understand the subsequent points made in this paragraph.

The paper claims that "the peaks have no intrinsic ordering," but I disagree.
Indeed, it seems quite strange to me that the peaks in the spectrum are not sorted by m/z for input to the model.

For the comparison to tokenized peak embedding (Section 2.1.1), it seems like a key parameter is the bin size of 0.1.  To make the comparison informative, you should vary this parameter to find an optimal setting for a given dataset.

The paper uses the term "molecular similarity," but I don't really know what this means.  For example, we are told that "Molecular similarity scores range from 0 to 1."  Which specific scores are referred to here, and how are they computed.  This seems critical because later there are thresholds (0.60 and 0.95) that get applied to these scores.

I find it highly problematic to report results based on an "internal, proprietary dataset," since this renders all of the results irreproducible.  The experiments should be performed only using publicly available data, and that data, including the splits, should be made available along with the paper.

Section 3.1 uses a lot of jargon that needs to be explained for an ICLR audience.  E.g., "positive and negative ion mode" and "collison energies."  What is involved in stripping stereochemistry from molecular structure labels, and why does this step allow for better molecule-disjoint splitting?  What are "chemical fingerprints"?

I don't understand what the phrase "lookup reference" refers to.  Why don't you just say that you query it against the training set?  What is being "looked up" here?  Later, you refer to a "reference set." Is this just the same as the training set?  Please be consistent in your terminology.

In the evaluation framework, why it it sensible to use arbitrary score thresholds (0.60 and 0.95)? Why not just use a rank-based approach akin to ROC?  It seems to me that if you really are interested in doing spectral library search, what matters is whether the top-ranked match for each query is in fact the correct molecule.  This gives you a ranked list of spectrum-spectrum matches, where each has a binary label of correct or incorrect.

What does it mean to "macro average" something?

I got lost in the paragraph that begins "Because novel molecules ..."  Where do the numbers 0.31 and 0.89 come from? Is this an empirical result, and if so, how is it computed.  I am not convinced that the random baseline is particularly interesting.  What does seem interesting is how the generalized scalar product, which is trivial to compute and requires no discretization, would perform.

I did not find the UMAP analysis particularly informative or compelling.

The property prediction task seems quite interesting, and I am surprised that no one has done this before.

Minor:

On p. 1, modify the sentence that talks about "modeling m/z values as numeric quantities" to clarify that you are contrasting this with the discretization approach.


**Summary Of The Paper:**

This paper proposes to use sinusoidal embeddings to represent m/z and intensity values when modeling metabolomics mass spectrometry data using deep learning models.  The idea is sensible though not particularly novel.  Such embeddings have been used previously for mass spectrometry data (as referenced in the related work section), just not for data generated from small molecules.  Three sets of results are given.  The first set I found confusing (see details below), but the upshot there seems to be that the proposed method works better than state-of-the-art methods.  The second set of results involve interpreting UMAPs.  I didn't find this section compelling, and I think it should be eliminated.  The third set of results involves predicting physical properties of molecules directly from spectra. This seemed like an interesting and potentially impactful application, and I was surprised that no one apparently has worked on this problem previously.


**Summary Of The Review:**

This paper addresses an important problem but uses a fairly straightforward technique that has been used previously in very similar settings.  The paper is at times hard to follow, and of the three sets of results only the third is very compelling.

---

> ### Author Response · Authors · 2022-11-19
> **Response to reviewer oTst part I**
>
> We thank the referee for their very helpful review. We hope our revisions make the story more clear. Responses to specific points are below.
>
> *In the paragraph that spans pp. 1-2, I don't understand why discretization techniques should be "particularly susceptible to measurement errors." Also, it seems strange to me that this discussion doesn't mention edge effects, which is another clear drawback of the discretization approach (separate from the loss of precision).*
>
> We have removed this claim. Also, we thank the referee for pointing out that we neglected to describe edge effects, and have added them to the manuscript.
>
> *The final paragraph of the introduction does not do a good job of making clear whether the proposed method represents an improvement relative to the current state of the art.*
>
> We have revised the introduction to include a clear statement of the contributions of this paper. In short, we show that models with sinusoidal embeddings as input create a new state of the art across multiple tasks, and that they do so by capturing high mass resolution information.
>
> *There is something important that I simply don't understand about the spectral library search application. My understanding is that the state of the art method is to use a sort of generalized scalar product (or cosine similarity) that is carried out directly on the non-discretized list of peaks. I think this is the method used in several widely used library search tools, e.g., from Pieter Dorrenstein's group. I think the paper should explain how methods like the ones cited in Section 1.1 (e.g., Huber 2021b) improve on this simple approach that does not require discretization. Results from a simple method like this, as well as perhaps cosine similarity carried out on discretized vectors, should be included in Table 1.*
>
> For spectral library search, it is indeed the case that a modified cosine similarity is the most commonly used approach in practice, including in the tools released by the Dorrestein group. Furthermore, as the referee points out, it does not require binning. We agree with the referee that including this standard method would strengthen the paper and have added it to the revised manuscript.
>
> Compared to cosine similarity, methods like those Huber 2021b have been reported to perform better in the literature, in spite of utilizing binning. Presumably the reason for this is that using a learned model for similarity, rather than an untrained similarity metric like cosine, is enough to lead to an improvement in spite of information loss and edge effects from binning. A key strength of our method is that it is both able to learn from the data without binning. We have added a brief discussion of this point to the manuscript in the conclusions section.
>
> *The paper claims that "the peaks have no intrinsic ordering," but I disagree. Indeed, it seems quite strange to me that the peaks in the spectrum are not sorted by m/z for input to the model.*
>
> Our view is that there are multiple possible orderings. For example, we could order by intensity. In our processing, the peaks are ordered by mass. However, our model, which uses no positional encodings, ultimately sees the peaks as a set of peaks and not a sequence.
>
> *For the comparison to tokenized peak embedding (Section 2.1.1), it seems like a key parameter is the bin size of 0.1. To make the comparison informative, you should vary this parameter to find an optimal setting for a given dataset.*
>
> We tried this at 0.01 Da bins for property prediction and found degraded of performance, possibly due to insufficient data to support the size of the resulting vocabulary. 1 Da bins are extremely lossy, as non-integer masses are one of the key analytical strengths of MS. We have not experimented with them, in line with existing literature.
>
> *The paper uses the term "molecular similarity," but I don't really know what this means. For example, we are told that "Molecular similarity scores range from 0 to 1." Which specific scores are referred to here, and how are they computed. This seems critical because later there are thresholds (0.60 and 0.95) that get applied to these scores.*
>
> We use tanimoto similarity computed from rdkit topological fingerprints for molecular similarity. For simplicity, we have dropped the threshold of 0.95 in the revised manuscript in favor of an exact match metric. The similarity threshold of 0.6 is used in the literature for analog search. In the revised manuscript we’ve briefly described this.

---

> ### Author Response · Authors · 2022-11-19
> **Response to reviewer oTst part II**
>
> *I find it highly problematic to report results based on an "internal, proprietary dataset," since this renders all of the results irreproducible. The experiments should be performed only using publicly available data, and that data, including the splits, should be made available along with the paper.*
>
> We have removed all proprietary data. The original manuscript only mentioned it for completeness, as it was .0015% of our data, just ~10 spectra hand annotated for other purposes that ended up in our dataset. Removing them did not alter our metrics at the resolution reported in the paper. Unfortunately, our dataset still contains commercially available data that we cannot share due to licensing agreements, but that are quite standard in publications in this field. For example, the NIST 2020 dataset is not public, but available for purchase at the price of 1570 USD. It would be a violation of our license to release them, but we believe it is valuable to the community to see what results look like over datasets that include standard sources like NIST that are only available with license and are of large scale. We have described our dataset construction and cited the source of every spectrum and structure. Additionally, we note that many groups in the field will already have possession of these datasets.
>
> *Section 3.1 uses a lot of jargon that needs to be explained for an ICLR audience. E.g., "positive and negative ion mode" and "collison energies." What is involved in stripping stereochemistry from molecular structure labels, and why does this step allow for better molecule-disjoint splitting? What are "chemical fingerprints"?*
>
> These are details included for the sake of reproducibility and have been moved to the reproducibility statement.
>
> *I don't understand what the phrase "lookup reference" refers to. Why don't you just say that you query it against the training set? What is being "looked up" here? Later, you refer to a "reference set." Is this just the same as the training set? Please be consistent in your terminology.*
>
> In the first paragraph of Section 3.2, we state that the lookup reference set is indeed our training set. We have clarified this section and have double checked our terminology for consistency.
>
> *In the evaluation framework, why it it sensible to use arbitrary score thresholds (0.60 and 0.95)? Why not just use a rank-based approach akin to ROC? It seems to me that if you really are interested in doing spectral library search, what matters is whether the top-ranked match for each query is in fact the correct molecule. This gives you a ranked list of spectrum-spectrum matches, where each has a binary label of correct or incorrect.*
>
> The score thresholds reflect two different goals. The goal of the threshold of 0.95 was to find essentially exact matches, but with a slight tolerance to, for example, a group that is fairly mobile and you see common interconversion. For simplicity, in the revised manuscript, we’ve adopted an exact match criterion and dropped the 0.95 results.
>
> The goal of the 0.6 threshold of molecular similarity is to capture how successfully the system finds structures that are similar, but not necessarily identical to the ground truth molecular structure. This threshold is used in the literature for this purpose and is important in applications where, for example, the ground truth molecule is not in a spectral library, but an approximation of the structure remains useful, as in drug screening applications where medicinal chemists would expect to modify any hit later anyway, and just want to know if the molecule is an appealing starting point.
>
> In the revised manuscript we adopt exact match in favor of 0.95, and describe the role of the 0.6 threshold for analog search in more detail.
>
> *What does it mean to "macro average" something?*
>
> Macro averaging is a standard method for reporting metrics when some classes (or structures in this case) have far more data representation than others. In MS/MS datasets, this is the case. Some structures have many spectra associated with them, while others have only a few, or one. Macro averaged results are generated by averaging over the performance for each of the spectra corresponding to an individual spectrum first, and then averaging those averages for the final metric value. As this is a standard terminology, and our original manuscript described that we do it over structures, we have not edited in response to this question.

---

> ### Author Response · Authors · 2022-11-19
> **Response to reviewer oTst part III**
>
> *I got lost in the paragraph that begins "Because novel molecules ..." Where do the numbers 0.31 and 0.89 come from? Is this an empirical result, and if so, how is it computed. I am not convinced that the random baseline is particularly interesting. What does seem interesting is how the generalized scalar product, which is trivial to compute and requires no discretization, would perform.*
>
> The purpose of this paragraph was to describe the performance of an oracle on the dataset. We have edited this section for comprehension.
>
> *The property prediction task seems quite interesting, and I am surprised that no one has done this before.*
>
> We thank the reviewer for this observation. We prefer to think of this as a set of several tasks.
>
> *On p. 1, modify the sentence that talks about "modeling m/z values as numeric quantities" to clarify that you are contrasting this with the discretization approach.*
>
> We have modified the manuscript as suggested.
>
> Summary of response:
> We find that most of this review’s content was suggestions on how to clarify our manuscript. We have made substantial revisions to attempt to clarify it, and have removed all proprietary data. Furthermore, we have clarified exactly what in our results we consider to be novel in the updated manuscript.
> We agree with the reviewer that it is regrettable that so much core data in metabolomics is only commercially available, but have provided citations to all sources so that labs can reproduce our data with appropriate licenses. We also have provided our code, and will release it upon publication.

---

### Official Review · Reviewer_EA9S · 2022-10-21

**Confidence:** 5
**Correctness:** 3
**Technical Novelty And Significance:** 1
**Empirical Novelty And Significance:** 2
**Recommendation:** 1

**Clarity, Quality, Novelty And Reproducibility:**

I don’t see any technical novelty in the paper. Sinusoidal embeddings were already proposed for the field of mass spectrometry (Yilmaz et al.) as noted by the authors. The idea of using neural networks for spectral library search is not novel either as confirmed by the authors. The results section is not easy to read if you are not very familiar with particular evaluation benchmarks in mass spectrometry in metabolomics.


**Strength And Weaknesses:**

The authors propose to use sinusoidal embedding to obtain a representation of the measured spectrum, allowing the comparison of similarities between measured compounds. The advantage of using sinusoidal embeddings in mass spectrometry is significant and was already demonstrated by Yilmaz et al. Here the authors additionally confirmed it with experiments in metabolomics.

Minor comments:
* What is a “development” set? What happened to the “validation” set? Why do the authors call
“validation” set “development” set?
* From the description in section 3.1 it is unclear whether test and validation sets can have an overlapping set of molecules. It is clear that they don’t overlap with training, but not clear if they overlap with each other. The whole section is not very clear. The part about known and not known molecules is not very clear either. Also stratification by spectrum alone and not by molecules is also unusual. I have concerns that the model can overfit individual molecules. The network proposed by the authors is much larger than the benchmarks and it is not clear to me how the authors demonstrated overfitting if the test set contains molecules present at training.
* Section 3.2 is not well written. The metrics are not well-defined. The thresholds seem arbitrary. It is not clear at all how these numbers were estimated. Was FDR used to find these cutoffs? Maybe these metrics are commonly used in metabolomics, but they should be clearly described for a machine learning conference.
* The authors demonstrate the improved numbers in their tables, but since the metrics and the whole evaluation are not transparent, it is hard to assess what is the actual significance of these numbers.



**Summary Of The Paper:**

The paper applied sinusoidal embeddings for spectral library search in mass spectrometry in metabolomics.

**Summary Of The Review:**

The paper empirically demonstrates the advantage of using sinusoidal embeddings for spectral library search in metabolomics. The idea of using sinusoidal embeddings to avoid binning of ms spectrum is not novel and was recently published. The authors take this idea one step further and demonstrate that it is superior that currently existing simple benchmarks for spectral library search in the field of metabolomics. The biggest contribution of this paper is the empirical demonstration of their results, but this section is not transparent at all.

---

> ### Author Response · Authors · 2022-11-19
> **Response to minor comments**
>
> 1. The referee wonders why we use the term development set instead of validation set. Development set is a commonly used synonym of validation set. The ubiquity of “development set” in machine learning is easy to verify from a string search of the proceedings of recent machine learning conferences and both terms are acceptable. We have a slight preference for the development set terminology because we find it more descriptive, and it avoids any possible confusion with test set (see the section on terminology confusion in https://en.wikipedia.org/wiki/Training,_validation,_and_test_data_sets ). We therefore do not alter our usage in the revised manuscript.
> 1. The referee asks if our development and test sets have molecular structure overlap with each other and with the training set and potential overfitting. The development and test sets are fully structure disjoint and spectrum disjoint. However, both the test and development sets contain a fixed fraction of structures (“known” in the manuscript) that are also contained in the training set. They do not contain any spectra that are contained in the training set. The reason for this unusual data split is that mass spectrometry for small molecules is not unique; many different spectra can come from the same molecule, depending on experimental conditions, a fact reflected in our dataset which contains large numbers of spectra for many individual molecules. Therefore, there are two ways of splitting the data, across spectra and across structures. Because in practical analysis of real metabolomics MS/MS experiments, identifying the structure and properties of molecules that are in spectral databases is a very important task (e.g. “dereplication”), we adopt this split. This allows us to report the performance of the models on novel spectra corresponding to structures both in and out of the training set. Since both of these metrics are of high practical importance, we included both in the manuscript. We are aware that this is an additional complexity in the data not common in many ML settings. To assist readers with a general ML background, we’ve revised the manuscript to clarify these decisions and the structure of the data.
> 1. The referee found section 3.2 not well written, and asked for clarification on the metrics chosen. Section 3.2 is focused on the task of spectral library search (SLS). This task uses a spectrum for an unknown molecule to search a database of spectra where the structure of the corresponding molecule is known. If a sufficiently similar spectrum is found, then the assumption is that the corresponding molecule is the unknown molecule, or if the similarity is lower, a structural analog. SLS research therefore focuses on learning similarity functions over spectra that correspond well with a desired notion of chemical structure similarity. The metrics we describe in section 3.2 describe the success rate of SLS in finding molecules that meet two different thresholds of structural similarity between the ground truth molecule and the molecule retrieved by SLS. Structural similarity is measured by the Jaccard (Tanimoto) similarity of standard rdkit topological molecular fingerprints. The threshold of 1.0 is an essentially perfect match, while the threshold of 0.6 is used in the literature as a threshold for two structures to be analogs. We’ve substantially edited this section to clarify the metrics used, their motivations, and added relevant citations for the analog metric.
> 1. Finally, the referee notes that because of the lack of clarity in the metrics, it’s hard to assess their significance. We hope that our clarifications for SLS above help with the SLS portion. For property prediction, we used standard regression metrics, and properties that are of common interest to medicinal chemists.

---

> ### Author Response · Authors · 2022-11-19
> **Response to clarity, quality, novelty and summary**
>
> The reviewer had two main concerns in these sections. First, the reviewer points to our citation of Yilmaz et al. that applies sinusoidal embeddings to Proteomics mass spectrometry in the context of de novo protein sequencing. We agree (as per our citation) that this paper made an important contribution. However, the Yilmaz paper is quite distinct from the present work in three respects. First, it is restricted to a single task in the field of proteomics. Proteomics considers vastly different data than small molecule chemistry and metabolomics. Proteins are 2-3 orders of magnitude higher mass than in metabolomics, and are sequential, while small molecules have a general graph structure (to first approximation). Because of this, proteomics is methodologically distinct from metabolomics. As reflected in the distinct literatures that have built up in both fields, it’s far from a given that methods will successfully transfer between the fields. Second, the Yilmaz paper simply adopts sinusoidal embeddings as one design decision in a larger model. They do not carry out any study of the impact of this decision, how it compares to alternatives, or consider whether any gains might carry across to multiple tasks. Beyond a short verbal argument, there is no evidence presented in that paper that the usage of sinusoidal embeddings benefits their model at all. None of this should be construed as criticism of the Yilmaz paper; these concerns were not the purpose of that paper. However, our manuscript does focus on these important questions, and characterizes sinusoidal embeddings in four critical ways:
> * We compare to discretization approaches that are common in the literature and show that sinusoidal embeddings outperform
> * We compare across numerical floating point precisions for the mass inputs, and show that sinusoidal embeddings perform better when higher mass resolution data is input, showing that, unlike standard binning approaches, neural networks with sinusoidal embeddings for the input masses are able to learn from high resolution mass data.
> * We show that performance improvements generalize across spectral library search for both analog and exact match molecules, and for 10 different property prediction regression tasks. This pattern of results across tasks provides additional evidence that sinusoidal embeddings are an effective
> * We show evidence that there is a high-dimensional structure to the mass embeddings that emerges only when sufficiently high precision mass data is input
> With this evidence taken together, we consider our manuscript to show the first persuasive evidence that mass spectrometry data can be effectively encoded using sinusoidal embeddings across tasks, and that the embeddings do indeed learn from the high mass resolution that is characteristic of modern mass spectrometry experiments.
>
> To communicate these contributions more clearly, we have rewritten our related work section to describe the Yilmaz et al. work more clearly, and have rewritten our conclusions and introduction sections to highlight clearly the contributions described in the bullet points above. In light of these contributions, we believe that our work is both correct, and sufficiently technically novel and important to merit acceptance at ICLR.

---

> > ### Comment · Reviewer_EA9S · 2022-11-19
> > **technical novelty**
> >
> > Dear authors,
> >
> > thank you for the detailed feedback. I don't doubt that the improvements you get compared to state-of-the-art methods in metabolomics are interesting, however, I still don't see any novelty from a machine-learning point of view and therefore have a concern how suitable this paper for ICLR. Also, yes, historically metabolomics and proteomics subfields don't overlap much, but it doesn't matter that the posed problem doesn't look similar from a machine-learning point of view. It is great that the straightforward application of sinusoidal embeddings worked so well for your task, but it is hard to see any challenge applied beyond applied machine learning. It is even harder to do so, given the datasets you use are not freely available.

---

> > > ### Author Response · Authors · 2022-12-03
> > > **re. technical novelty**
> > >
> > > The feedback from reviewer EA9S is primarily focused on lack of novelty due to the related work by Yilmaz et al. 2022. While we explain how our work differs from Yilmaz et al. in our updated manuscript and in our reply to reviewer EA9S, we would also like to highlight that according to ICLR FAQ for Reviewers (https://iclr.cc/Conferences/2023/ReviewerGuide#FAQ) a work is considered contemporaneous if it was published in a peer-reviewed journal on or after May 28, 2022. The work in question, Yilmaz et al. 2022, was presented at ICML 2022 which took place over 17 - 23 July 2022. As the papers are contemporaneous, we weren’t required to discuss it at all and we don’t think the novelty assessment of our paper should be impacted, especially given the different focuses of the work. We note that we discovered the Yilmaz paper while preparing our manuscript for publication.
> > >
> > > As discussed in our response to other reviewers, the full datasets indeed cannot be publicly released, but that is due to third party licenses that we do not control. The licensed datasets are standard in metabolomics labs and are commercially available.  It’s regrettable that these datasets are not open, however they comprise a large fraction of the appropriate data, and ignoring them would reduce the utility of this publication to the community of scientists in the area that use them routinely. The code will be released upon publication.
> > >
> > > We also want to note that we disagree with the referee EA9S on the suitability of this paper for ICLR. In our view, applying ML to MS/MS has been historically hampered by the lack of an effective and efficient method for representing high mass resolution data (parts per million). Our paper goes well beyond trying sinusoidal embeddings with an MLP and showing that they work, but rather shows, among other things, that the resulting models are responsive to the high resolution portion of MS/MS data. Such work falls well inside ICLRs topics of interest as outlined in the [call for papers](https://iclr.cc/Conferences/2023/CallForPapers), such as “applications in . . . biology, or any other field”, “learning representations of outputs or states”, and “visualization or interpretation of learned representations”.

---

### Decision · Program_Chairs · 2023-01-20

**Decision:**

Reject

**Justification For Why Not Higher Score:**

The topic is a bit distant from ICLR and the writing is unclear.

**Justification For Why Not Lower Score:**

N/A

**Metareview: Summary, Strengths And Weaknesses:**

The paper proposes multi-scale sinusoidal embeddings for spectral library search.

Strength of the paper:
1. The idea of sinusoidal embeddings seems to be well-motivated for this special task: Mass Spectrometry analysis.

Weakness of the paper:
1. The current writing is inaccessible to ICLR audience with a lot of confusing expressions.